# Analysis of Potential Evapotranspiration in Heilongjiang Province

**Quanchong Su** [1,2], **Changlei Dai** [1,2,*], **Qingsong Zhang** [1,3] **and Yang Zhou** [1,2]

[1] Institute of Groundwater in Cold Region, Heilongjiang University, Harbin 150080, China; hss_suquanchongi@163.com (Q.S.); zqs1714011041@163.com (Q.Z.); hss_zhouyang@126.com (Y.Z.)
[2] School of Hydraulic and Electric Power, Heilongjiang University, Harbin 150080, China
[3] Northeast Institute of Geography and Agroecology, Chinese Academy of Sciences, Changchun 130102, China
[*] Correspondence: daichanglei@126.com; Tel.: +86-0451-8660-4008

**Abstract:** As the global temperature has been increasing, an intriguing phenomenon has emerged wherein the potential evapotranspiration (PET) in many areas does not show a clear positive correlation with the temperature. Instead, PET is decreasing in various areas, giving rise to the "evaporation paradox". This phenomenon also occurs in Heilongjiang province. Heilongjiang province is a province in China with a large agricultural output, so it is necessary to explain the mechanism of decreasing pan evaporation in order to cope with the rational use of food and water resources in China. In this paper, the Mann–Kendall (M-K) parameter test and multiple regression analysis were used to analyze the trends and correlation between PET and climate change in Heilongjiang province from 1961–2020. The results show that the potential evaporation in Heilongjiang province had a decreasing trend of 7.776 mm/10 a, and there was a fluctuation in the decreasing trend during 1985–1995 and 2000–2020. Temperature, wind speed, vapor pressure, and solar radiation contributed $-87.23\%$, $88.25\%$, $65.87\%$, and $42.53\%$ to PET variation, respectively, highlighting wind speed as the main factor in the decrease in PET, followed by the vapor pressure and solar radiation, whereas temperature-induced changes in PET were neutralized by the former.

**Keywords:** potential evapotranspiration; multiple regression analysis; M–K test; evaporation paradox; Heilongjiang province

## 1. Introduction

The hydrological cycle includes evaporation, precipitation, and runoff. Among them, evaporation is the process of transforming solid or gaseous states into a liquid state [1], which is an important component of the water cycle. Typically, the main factor affecting the magnitude of evaporation is temperature, and an increase in temperature will lead to a corresponding increase in evaporation [2]. However, recent studies have shown that as global temperatures rise, evaporation does not show the expected changes; instead, pan evaporation continues to decrease in many areas. For example, in 1995, Peterson et al. [3] pointed out a decline in global total evaporation despite a 0.13 °C temperature rise per decade over the past 50 years, coining this paradoxical situation the "evaporation paradox". Accordingly, this phenomenon emerged as the origin of the evapotranspiration paradox theory [4]. Although exceptions have been made in individual regions, the evaporation paradox has been evidenced in most regions worldwide, such as Italy [5], India [6], Australia [7], Israel [8], New Zealand [9], and China [10–12]. Since then, many scholars domestically and internationally have studied the causes of the evaporation paradox and the related problems it may bring about. However, there is no clear explanation of the formation mechanism of the evaporation paradox, and most studies are qualitative analyses or analyses combined with relevant detection data, which is also closely related to the many factors affecting evaporation. This complexity stems from numerous factors affecting

evaporation apart from temperature, including wind speed [13], relative humidity [14], average atmospheric pressure [15], geographical environment [16], and solar radiation [17].

For different regions, there are many methods to analyze the generation of this phenomenon. Palmer [18] proposed a temperature-based method to estimate potential evapotranspiration (PET) in correlation with rising global temperatures. Despite its widespread use, this approach is fundamentally flawed. In particular, contemporaneous studies have highlighted the shortcomings of temperature-based methods for estimating PET when used for climate change assessments. In 2012, a special report by Seneviratne et al. [19] on extreme weather conditions emphasized and discussed the temperature-based approach to calculating evapotranspiration, indicating that while one should focus on temperature, one should also use a physical approach (a function of radiation, temperature, humidity, and wind speed) for calculating PET, proposing a new flux-based framework for measuring prolonged droughts and then predicting global evapotranspiration paradoxes using the framework and the same climate modeling investigations. The framework recognizes meteorological, hydrological, and agroecological perspectives, and explicitly includes the biological impacts of $CO_2$ change. The researchers also proposed the well-known drought index, which is based on the correlation of spatial changes in long-term mean precipitation (P) and temperature (T) with the spatial distribution of major vegetation types. Maliva and Missimer [20] outlined the historical development of this field. Thornthwaite [21] introduced the concept of PET (evaporation). In addition, the evaporation rate of class A evaporation panes at many observatories in Australia decreased during 1970–2002 [22], and the evaporation rate decreased at the same time as the temperature increased. Using the Penman evaporation model, He et al. [23] attributed the changing trend of evaporation in Australia from 1975–2004 to the influence of other climate change factors and believed that the changing trend of daily average wind speed was an important factor affecting the changing trend of evaporation. At the same time, strong climate change can also affect the transport of sediment within the catchment, thereby altering the local topography and hydrological cycles [24–26], especially on steep slopes with dense vegetation and wetlands [27].

Heilongjiang province contains a large amount of black soil and is the largest grain-producing area in China [28], with the province's Sanjiang Plain and Songnen Plain, two of China's nine major grain-producing bases. The study of climate change and evaporation change trends in this region are of great significance to China's grain engineering [29]. In the past 10 years, the temperature in Heilongjiang province has shown a fluctuating upward trend (in the past 100 years, the temperature has increased by about 0.6–0.9 °C) [30]. In recent years, most scholars have studied climate change in Heilongjiang province, mainly focusing on the influence [31,32] of crop growth and grain yield. There is a lack of quantitative analysis of the contribution rate of meteorological factors to pan evaporation. The purpose of this paper is to study the potential evaporation and climate change trends in Heilongjiang province, quantitatively analyze the contribution rates of temperature, wind speed, solar radiation, and air pressure to the potential evaporation change, judge the action form of the evaporation paradox in this region, predict the regional water budget under global warming, and take relevant effective measures to reduce the negative impact of environmental change.

## 2. Study Area and Data Processing

As the northernmost and easternmost province in China, Heilongjiang province is across the Amur River from Russia to the north and east, adjacent to the Inner Mongolia Autonomous Region to the west, and bordering Jilin province to the south. It is located between $121°11'-135°05'$ E and $43°26'-53°33'$ N, with a total area of 473,000 km$^2$ [33] (Figure 1). With the influence of global warming, the overall temperature change trend in Heilongjiang province in recent years is consistent with the overall change. The average temperature shows an upward trend, and the temperature difference decreases year by year. In winter, the temperature rises faster [34], and precipitation and snowfall both increase

to varying degrees [35]. The trend rates for relative humidity and sunshine duration decreased year by year at −0.32 h/10 a and −32.4 h/10 a, respectively [36]. Climate change in Heilongjiang province has a significant impact on the environment. According to research, the permafrost areas in Heilongjiang province have been degraded to varying degrees due to the rising temperature, with the Da Hinggan Mountain region being the most significant [37].

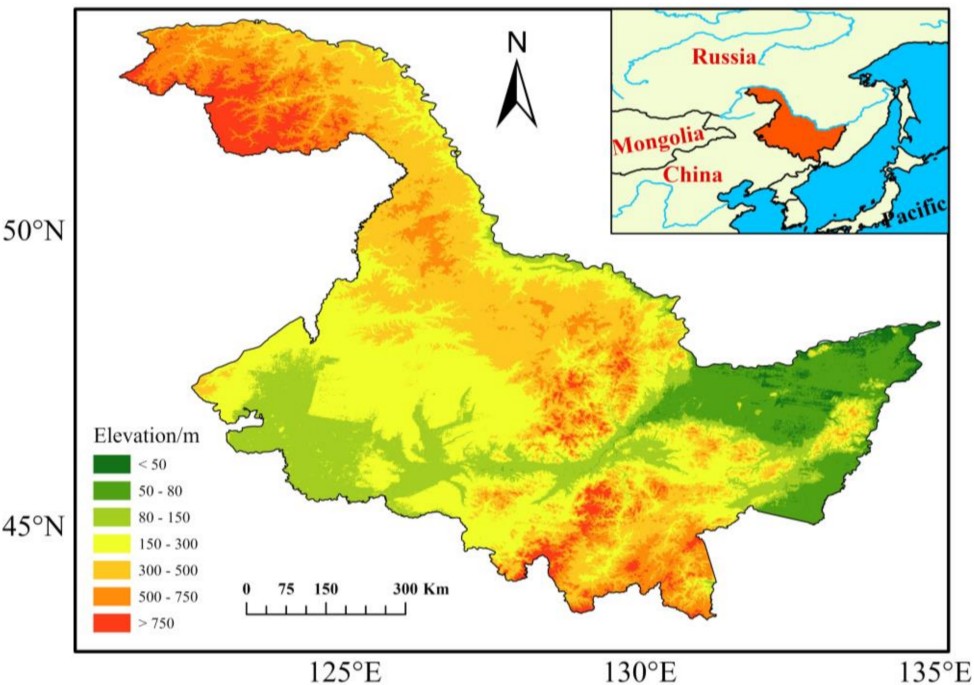

**Figure 1.** Geographical location of Heilongjiang province.

The evaporation data used in this paper are from the monthly PET in China from 1961 to 2020, provided by the National Tibetan Plateau Scientific Data Center (https://data.tpdc.ac.cn/zh-hans/data/8b11da09-1a40-4014-bd3d-2b86e6dccad4/?q=) (accessed on 15 August 2023)). The data set was based on the 1 km monthly mean temperature, minimum temperature, and maximum temperature data set for China [38], and the PET calculation formula was used to obtain the data. Meteorological data are from the National Tibetan Plateau Scientific Data Center (https://data.tpdc.ac.cn/zh-hans/data/99dd84e2-288d-4db8-b098-8118b3b0c17a) (accessed on 10 August 2023) in 1983–2017. The dataset is based on the existing international Princeton reanalysis data, GLDAS data, GEWEX-SRB radiation data, and TRMM precipitation data as the background field, with the routine meteorological observation data of the China Meteorological Administration integrated.

## 3. Materials and Methods

### 3.1. Trend Analysis Using the Mann–Kendall (M–K) Test

The rank-based nonparametric M–K test is a climate diagnosis and prediction technique [39,40], which is widely used to analyze the trend of long time-series data [41,42]. This method proves especially effective in identifying abrupt changes within climate series and determining the specific time point of such shifts. The M–K test is also often used to detect trends of precipitation and drought frequency under the influence of climate change, which does not require the samples to follow a certain distribution, is not disturbed by a few outliers, is more suitable for type variables and sequential variables, and has a

relatively simple calculation. Firstly, for a time series X with a sample size of n, the order column is constructed as follows:

$$S_k = \sum_{i=1}^{k} r_i \qquad r_i = \begin{cases} 1 & x_i > x_j \\ 0 & else \end{cases} \tag{1}$$

where order column $S_k$ is the cumulative number of values at time $i$ greater than the number of values at time $j$. Under the assumption of random independence of the time series, the statistics can be defined as follows:

$$UF_K = \frac{S_k - E(S_k)}{\sqrt{Var(S_k)}} \qquad k = 1, 2, 3, \ldots, n \tag{2}$$

where $UF_1 = 0$, $E(S_k)$, and $Var(S_k)$ are the mean and variance of the cumulative number $S_k$, which is independent in $x_1, x_2, \ldots, x_n$ of each other with the same continuous distribution. It can be calculated as follows:

$$VarS_k = \frac{n(n-1)(2n+5)}{72} \tag{3}$$

where $UF_i$ is the standard normal distribution, calculated in statistical sequence by the time series $x$ order $x_1, x_2, \ldots, x_n$. By referencing the normal distribution table with a predetermined significance level $\alpha$, the condition $|UF_i| > Ua$ indicates a clear trend change in the sequence. The M–K test was used to analyze the trends in evaporation, temperature, wind speed, and vapor pressure in Heilongjiang province. At the significance level of $\alpha = 0.05$, the M–K statistic $UF_i > 1.96$ indicates a significant increase, and $Z < -1.96$ indicates a significant decrease.

Linear regression was used to analyze the relationship between time and each variable, and a linear regression equation was established as follows:

$$x_i = at_i + b \tag{4}$$

where a is the regression coefficient, whether positive or negative, and signifies the trend of the meteorological factor increasing or decreasing over time, and b is the regression constant. The fitting parameters estimated by least squares are a and b. The calculation equation is expressed as follows:

$$a = \frac{n \sum x_i y_i - \sum x_i y_i}{n \sum x^2 - (\sum x_i)^2}$$

$$b = \frac{\sum y_i}{n} - a \frac{\sum x_i}{n} \tag{5}$$

Additionally, the distance horizon method can be utilized to determine the degree of variability in a long time series. The distance-parity value, which is used to determine whether the data for a certain period are relatively high or low in relation to a certain long-term average (e.g., a 30-year average) of the data, is a commonly used method to deal with the relationship between meteorological data and temporal variations. The original value is generally used to reflect the actual level of a particular period. Cumulative distance level represents the accumulation of distance levels. The main step is to first define a set of variables $(x_1, x_2, \ldots x_n)$, calculate the average value $\overline{x}$, and subtract each variable from $\overline{x}$, namely $x_i - \overline{x}$. The corresponding time of the variable is $t_i$. Find the sum of all the distance flat values before $t_i$, and $\sum_{i=1}^{t}(x_i - \overline{x})$ is the cumulative distance. Then, drawing the cumulative anomaly and time-change curve through the fluctuations of the curve change can judge the long-term evolution of the upward or downward trend.

*3.2. Analysis of Trend Causes*

Multiple regression equations were established by taking the main corresponding factor as a variable with the evaporation volume of the pan evaporation. Multiple regression analysis is a mathematical model of linear or non-linear relationships between multiple variables by making one variable a cause and another one or more variables independent variables. Furthermore, the linear correlation between multiple independent variables and multiple dependent variables is also investigated, called a multiple-multiple regression analysis model (or simply a many-to-many regression). Its mathematical form is as follows:

$$Y = \beta_0 + \beta_1 x_1 + \beta_2 x_2 + \ldots + \beta_n n \tag{6}$$

For the evaporation volume of the pan evaporation, it is generally believed that it has a good linear relationship with the reference evaporation, as follows:

$$E_{pan} = K_p ET_{ref} + K_c \tag{7}$$

where $K_p$ and $K_c$ are regression parameters, $E_{pan}$ is the evaporation volume of the pan evaporation, and $T_{ref}$ is the reference evaporation. The reference evaporation is the Penman–Monteith model revised by the FAO in 1998 to calculate PET [43,44], with the following equation:

$$ET_0 = \frac{0.408\Delta(R_n - G) + \gamma \frac{900}{T+273} U_2(e_s - e_a)}{\Delta + \gamma(1 + 0.34U_2)} \tag{8}$$

where $R_n$ is the net surface radiation of the reference crop in MJ/(m²· d); G is the soil heat flux in MJ/(m²· d); $\gamma$ represents the dry and wet coefficients in kPa/°C; $U_2$ is the wind speed at a height of 2 m on the surface in m/s; $e_s$ is the average saturated water vapor pressure in kPa; $e_a$ is the average saturated water vapor pressure in kPa; and T is the mean air temperature in °C.

Subsequently, the radiation term is parametrically corrected to be more general, and the empirical formula for calculating the radiation coefficient is as follows:

$$R_n = (1 - \alpha)\left(0.25 + 0.5\frac{n}{N}\right)R_{so} - \sigma\left(T_{x,k}^4 + T_{n,k}^4\right)\left(0.34 - \sqrt{e_a}\left(0.1 + 0.9\frac{n}{N}\right)\right) \tag{9}$$

where $\sigma$ is the Stefan–Boltzmann constant ($4.903 \times 10^{-9}$ MJ/K4·m²· d); $T_{x,k}$ and $T_{n,k}$ are the maximum and minimum temperatures of the absolute temperature scale in K; $n$ is the actual sunshine time in *h*; $N$ is the illumination time in *h*; $R_{so}$ is the radiation in sunny days in MJ/m²; $a_s$ is the component of the surface radiation under all cover ($n = 0$); $b_s$ is the component of the radiation from the atmosphere to the ground in sunny days (n = N); and $\alpha$ is the surface reflectivity (0.23).

According to the calculation relationship between PET rate and each meteorological factor of Equation (8), the calculation equation for the contribution to the PET rate can be obtained as follows:

$$\frac{dET_{ref}}{dt} = \frac{\partial ET_{ref}}{\partial R_n}\frac{dR_n}{dt} + \frac{\partial ET_{ref}}{\partial T}\frac{dT}{dt} + \frac{\partial ET_{ref}}{\partial U_2}\frac{dU_2}{dt} + \frac{\partial ET_{ref}}{\partial e_a}\frac{de_a}{dt} \tag{10}$$

## 4. Results and Discussion

*4.1. PET Trends*

Using Matlab (R2019a) to input the M–K test code, the evaporation data from 30 meteorological stations within Heilongjiang province were loaded (the site information is shown in Table 1), and the trend of PET at each meteorological station was derived, as shown in Figure 2. The change in evaporation in Heilongjiang province as a whole was not significant, with a non-significant decrease at 20 stations and a non-significant increase at 10 stations. Using the linear regression equation, Heilongjiang province PET as a whole

showed a fluctuating downward trend, with a decrease of 7.776 mm/10 a, as shown in Figure 3; the average PET from 1961 to 2020 was 775.6 mm, with a maximum value of 841.2 mm, a minimum value of 744.56 mm, and a downward slope of −0.57823. However, there is still an increase in evaporation within part of the time interval, as shown in Figure 4, according to the distance and cumulative distance plots.

**Table 1.** Annual change rate of each meteorological factor.

| Year | T-Min (°C) | T-Mean (°C) | T-Max (°C) | U (m/s²) | VP (%) | Rs (w/m²) |
|---|---|---|---|---|---|---|
| 1985–1994 | −3.288 | 2.412 | 8.117 | 3.011 | 0.663 | 15.011 |
| 1995–2004 | −2.859 | 2.862 | 8.55 | 2.899 | 0.656 | 15.141 |
| 2005–2014 | −3.019 | 2.596 | 8.208 | 2.606 | 0.667 | 15.017 |
| Speed/10 a | 0.1345 | 0.092 | 0.0455 | −0.2025 | 0.002 | −0.003 |
| Slope | 0.01959 | 0.01695 | 0.01482 | −0.01651 | 0.00038 | −0.00799 |

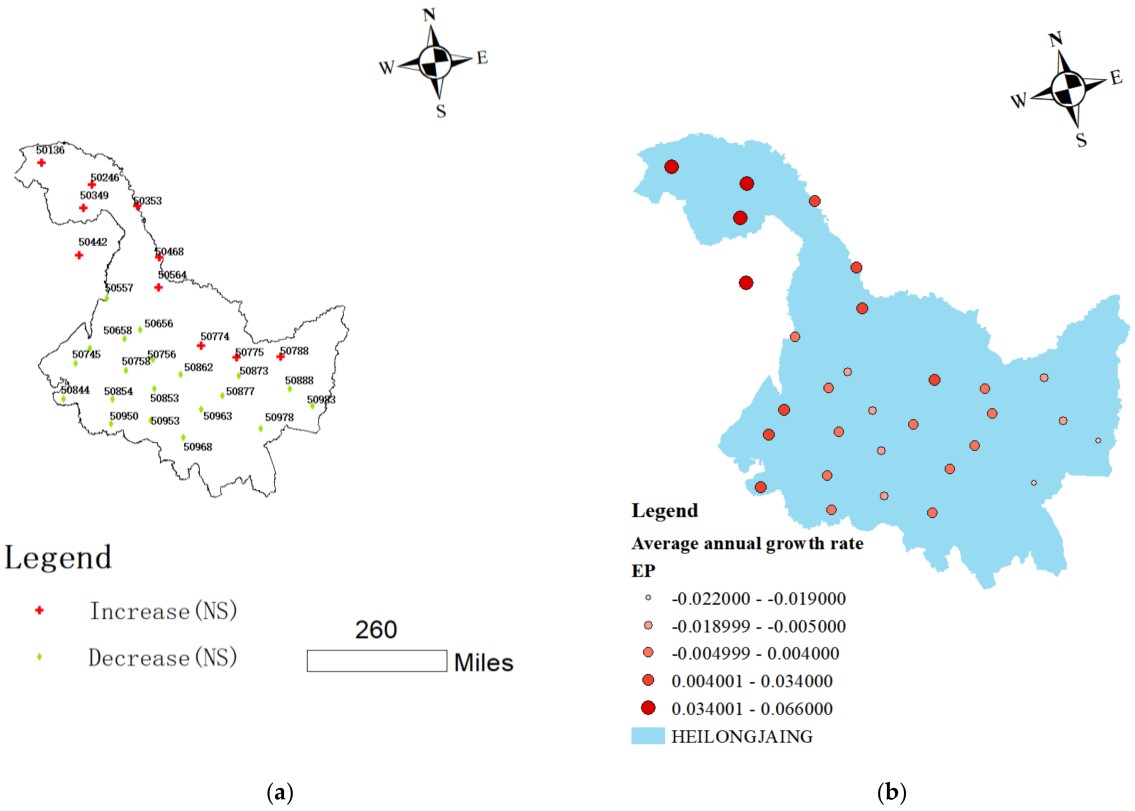

(**a**)                    (**b**)

**Figure 2.** PET trends at 30 meteorological stations in Heilongjiang province. (**a**) Degree of significant change; (**b**) rate of increase.

The intervals of 1961–1965, 1975–1980, and 1992–2002 showed an increasing PET trend, while the rest of the years showed a decreasing trend. The number of positive and negative alternations in the distance plot is as high as 19 times, indicating that PET fluctuates greatly. Seasonal PET was divided by meteorological seasons [45], March–May in spring, June–August in summer, September–November in autumn, and December–February in winter, and the trend of PET year by year in the four seasons of Heilongjiang province was analyzed, as shown in Figure 5.

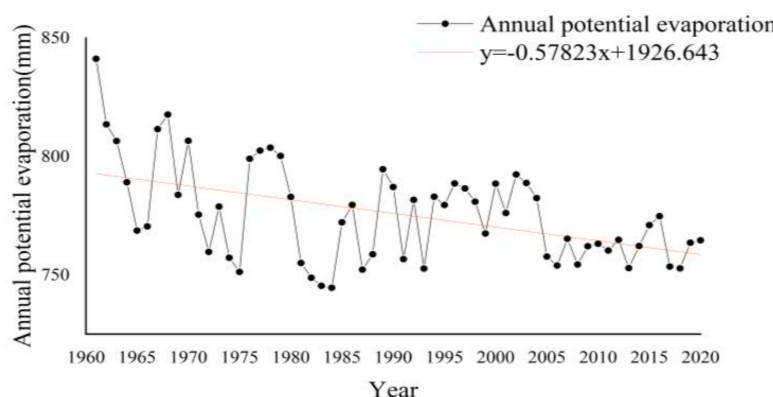

**Figure 3.** Interannual variation of PET in Heilongjiang province.

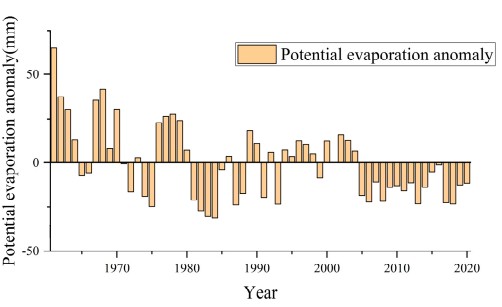

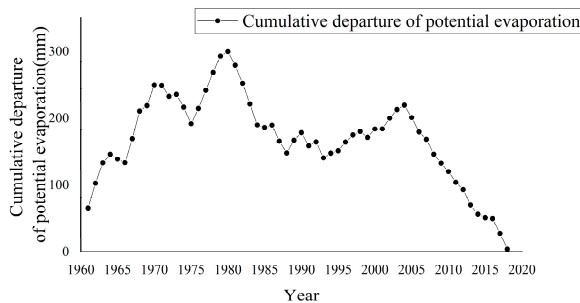

**Figure 4.** Anomaly and cumulative anomaly of PET.

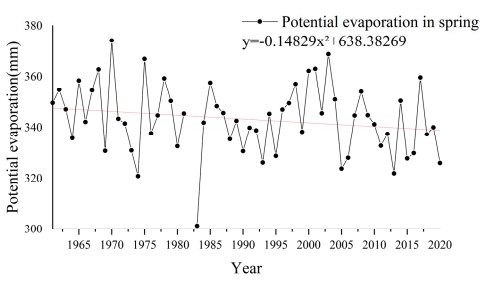

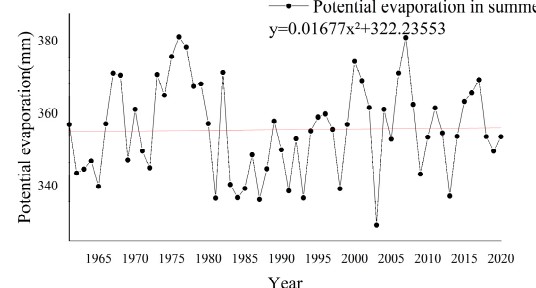

(**a**)

(**b**)

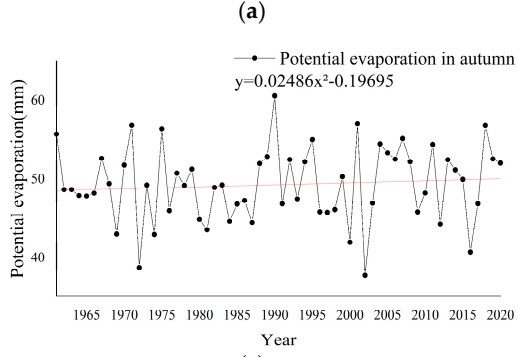

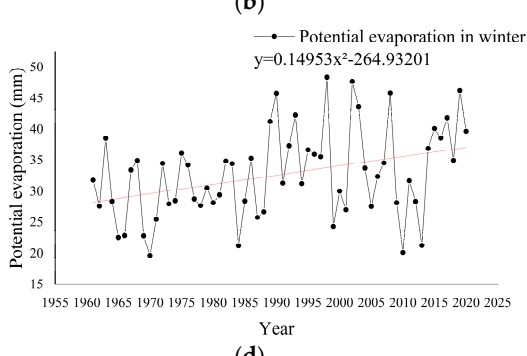

(**c**)

(**d**)

**Figure 5.** PET trends in the four seasons of Heilongjiang province. (**a**) spring; (**b**) summer; (**c**) autumn); (**d**) winter.

Through Figure 6, it can be seen that the two straight PET lines in spring in Heilongjiang province, UF and UB, were all over the critical range, which indicated that PET showed a non-significant upward trend. The PET trend in summer 1976–1980 was 2.934 mm/10 a; there was a significant temporal mutation in PET, and the mean value of

evaporation before and after the mean value of 370 mm was 350 mm; in fall, the UF line alternated more positively and negatively but was within the critical range, indicating that the change was not significant, and the whole was a non-significant upward trend; winter was similar to summer, and the UF line exceeded the critical value many times after 1995; the Heilongjiang province PET was 3.5 mm/10 a. Winter was similar to summer; the UF line exceeded the critical value several times after 1995, there are obvious changes, but the overall trend upward is not significant; there are several abrupt points within 1984–2014.

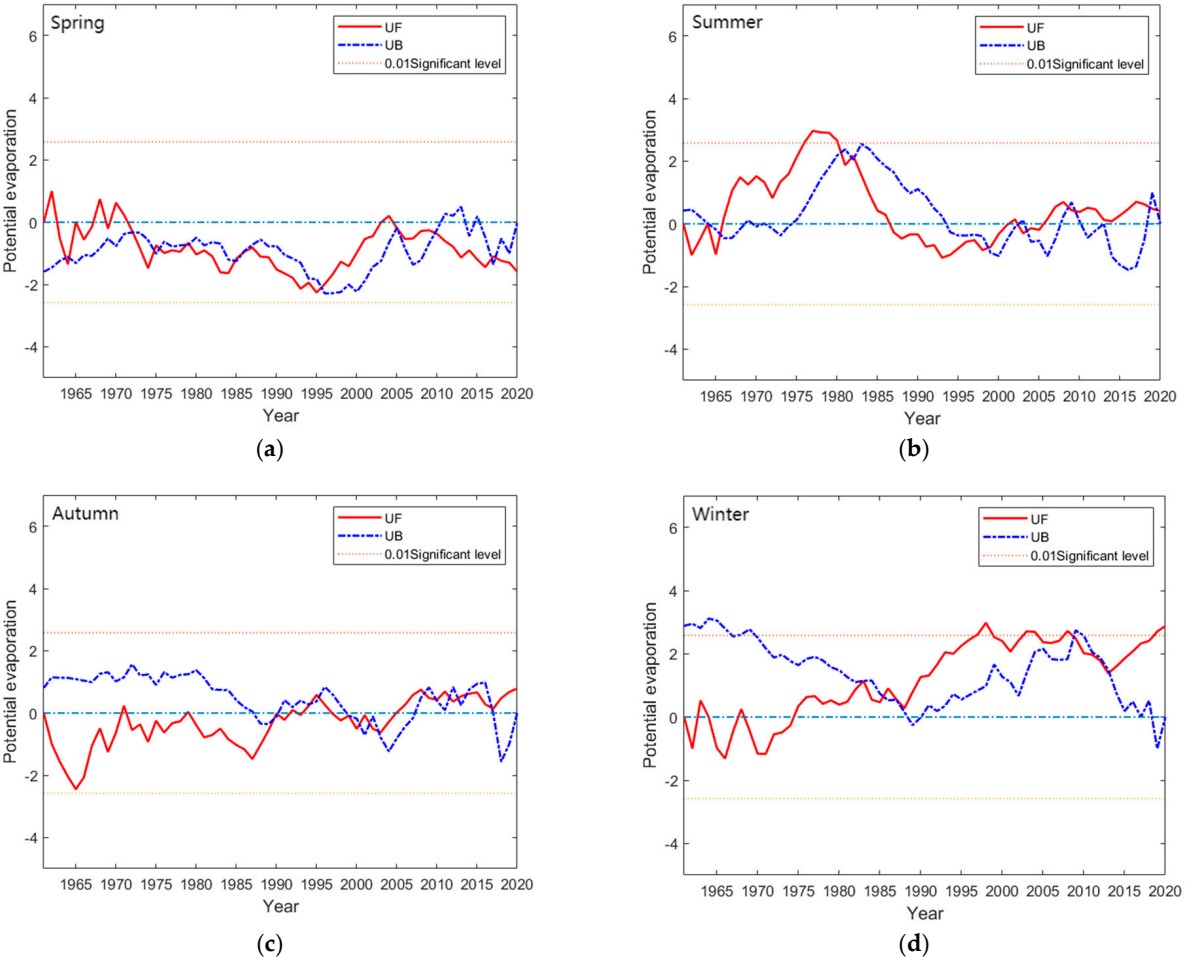

**Figure 6.** M–K test of four-season changes in PET in Heilongjiang province; (**a**) spring; (**b**) summer; (**c**) autumn); (**d**) winter.

The Kriging interpolation method was used to interpolate the average annual PET of 30 meteorological stations in Heilongjiang province, and the spatial distribution is shown in Figure 7. The mean error was −0.047 and the root mean square 1.357. It can be seen that PET changes gradually increase from north to south, with the minimum values occurring in the Da Hinggan Mountains and Xiao Hinggan Mountains as well as in areas of Mudanjiang, with the maximum value appearing in the southern Sanjiang Plain.

*4.2. Trends in Meteorological Factors*

In this paper, the maximum, minimum, and average temperature, vapor pressure, wind speed, and solar radiation data from 30 weather stations across Heilongjiang province from 1984 to 2017 were analyzed. Figure 8 shows the changing trends for each meteorological factor. In general, the temperature from the 30 meteorological stations showed an upward trend, the maximum temperature of 5 stations showed a significant upward trend, and the rest showed an insignificant upward trend. The average temperature and the minimum temperature both showed a significant upward trend. The wind speed

showed a decreasing trend in general; 28 sites showed a significantly decreasing trend and 2 sites showed an insignificant decreasing trend. The vapor pressure showed an overall upward trend; 6 sites showed a significant upward trend, while the other sites showed an insignificant upward trend. The overall change in solar radiation is more complex; 15 stations showed no significant increase trend, 15 stations showed no significant decrease trend, and most of the meteorological factors changed significantly in high-latitude meteorological stations.

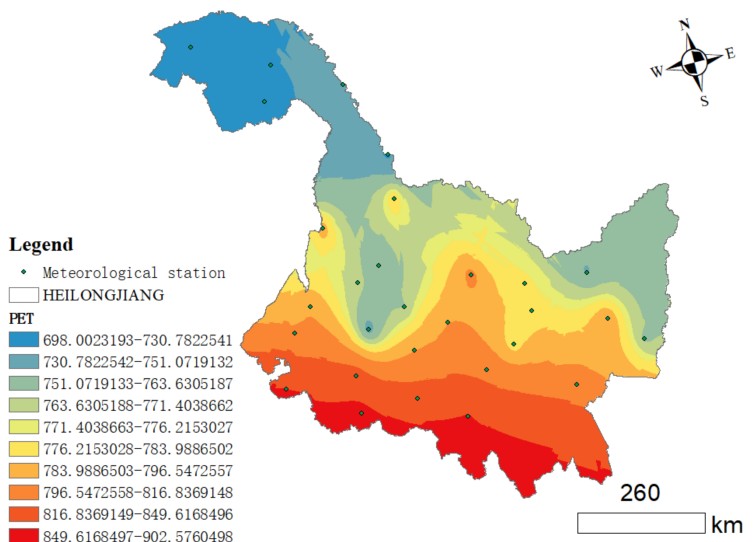

**Figure 7.** Spatial variation of PET in Heilongjiang province.

Figure 9 shows the interannual variation trends for various meteorological factors. The maximum temperature, average temperature, and minimum temperature increased at the rate of 0.0455 °C/10 a, 0.092 °C/10 a, and 0.1345 °C/10 a, respectively. The vapor pressure increased at a rate of 0.002 kPa/10 a, while wind speed and solar radiation decreased at a rate of 0.2025 $m/s^2 \cdot a^{-2}$ and 0.003 $w/m^2 \cdot a^{-2}$, respectively (Table 1). Multiple regression equations were established between PET and each meteorological factor (Figure 10), and it can be seen that each meteorological factor has a high correlation with PET, with a correlation coefficient of $R^2 = 0.814$.

The contribution of each meteorological factor to PET was calculated by Equation (10) (Table 2), which shows that temperature, wind speed, and PET were positively correlated; the correlation coefficient of wind speed was the highest at 1.9, indicating that changes in wind speed had the greatest effect on PET. Vapor pressure and solar radiation were negatively correlated to PET, with vapor pressure having the greatest effect, followed by solar radiation. The contribution of temperature, wind speed, vapor pressure, and solar radiation to the potential volume was −87.23%, 88.25%, 65.87%, and 42.53%, respectively, and although the effect of temperature on PET was also significant, the response results were gradually offset by wind speed, vapor pressure, and solar radiation.

**Table 2.** Regression analysis of PET and meteorological factors.

| Meteorological Factor | Intercept | Standard Error | T | P | Contribution Rate (%) |
|---|---|---|---|---|---|
| T-min | −79.38 | 0.013 | 1.2 | 0.02 | |
| T-mean | 79.02 | 0.031 | 1.6 | 0.03 | −87.23 |
| T-max | 16.52 | 0.022 | 1.5 | 0.01 | |
| U | −13.72 | 0.017 | 1.9 | 0.01 | 88.25 |
| VP | 238.58 | 0.063 | −1.7 | 0.06 | 65.87 |
| Rs | 12.14 | 0.092 | −1.5 | 0.15 | 42.53 |

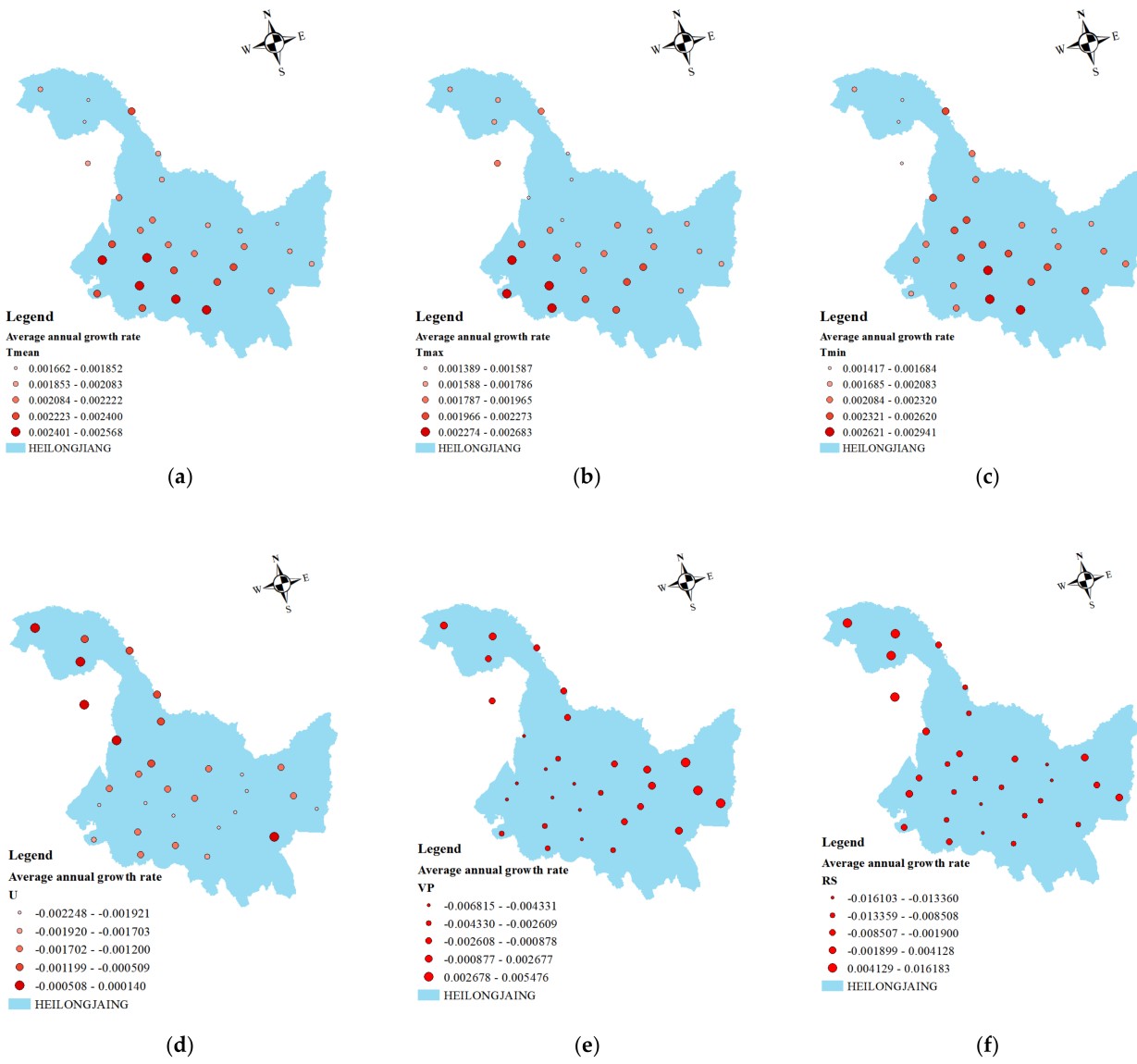

**Figure 8.** The changing trends of meteorological factors. (**a**) Mean temperature; (**b**) maximum temperature; (**c**) minimum temperature; (**d**) wind speed; (**e**) vapor pressure; (**f**) solar radiation.

PET is the main corresponding unit of global climate change. It is of great importance to study the trend of PET change under climate change in Heilongjiang province for the rational distribution of water resources and agricultural production in this region. The results showed that there was a decreasing trend in PET in Heilongjiang province, but the overall trend was not significant. The decreasing trend of PET in the Qinghai–Tibet Plateau, northwest, and southwest regions of China is very significant [46,47], indicating that it is not only related to meteorological factors but also to the geographical location of the study area and human activities. The results show that the variation in evaporation is the result of multiple environmental factors, and there are some regional differences in the correlation. This study shows that the variation in the trend of annual and seasonal evaporation is consistent with that of wind speed, which is basically consistent with existing research conclusions [48,49]. The spatial distribution characteristics of annual and seasonal evaporation trends are generally consistent with the research conclusions of Zhu Hongrui et al. [50]. However, there is a big difference from the research conclusions of Zeng Yan et al. and Qi Tianyao et al. [51–53]. The possible reason for this difference is that the time series of the research data are different. The data periods used in the above literature were 1960–2000 and 1960–2005, respectively, while the period of data in this study was 1961–2020.

As a result, the above literature is inconsistent with the conclusions of this study regarding the trends and periodic characteristics of the factors.

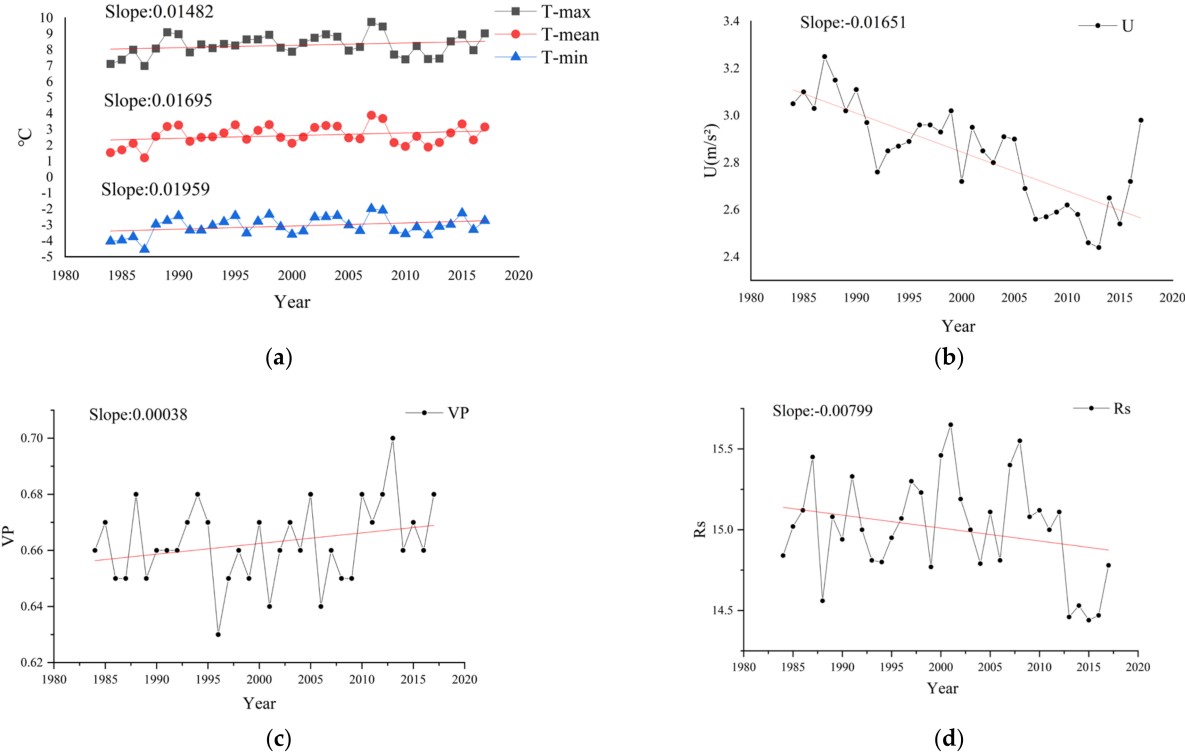

**Figure 9.** Interannual change trends for various meteorological factors. (**a**) Temperature; (**b**) wind speed; (**c**) vapor pressure; (**d**) solar radiation.

In addition, wind speed, as the main factor affecting the change in PET in Heilongjiang province, also has some corresponding relationship with the change in solar radiation. In areas where the sun shines for a long time, such as between the Tropic of Cancer and lower latitudes, the radiation is greater and therefore the temperature is higher, while the opposite is true in the middle and high latitudes. This difference in temperature caused by different amounts of solar radiation in low and high latitudes creates a pressure gradient that increases wind speed. At the same time, a large amount of greenhouse gas emissions will also lead to changes in solar radiation [54], but the corresponding relationship between the hydrological cycle and greenhouse gas emissions caused by human activities remains to be further studied.

Additionally, this paper mainly studied the relationship between the PET trend and various meteorological factors. However, changes in actual evaporation were not taken into account. The change in actual evaporation is not only affected by PET [55,56], but is mainly judged by the drought index. The drought index is the ratio of actual evaporation to precipitation, which can reflect the degree of dryness and wetness of a region. When the drought index is less than 0.8, the actual evaporation is positively correlated with PET. When the drought index is greater than 0.8 and less than 1, it is still a humid area, and the actual evaporation is affected by multiple climatic factors as well as precipitation, with meteorological factors accounting for a large proportion. However, when the drought index is greater than 1, it is already a completely arid area, and the actual evaporation and PET are obviously complementary [57]; the greater the drought index is, the greater the impact of actual evaporation on precipitation [58]. Therefore, in the follow-up study, the relationship between actual evapotranspiration and PET in Heilongjiang province will be further explored and its influencing factors will be analyzed.

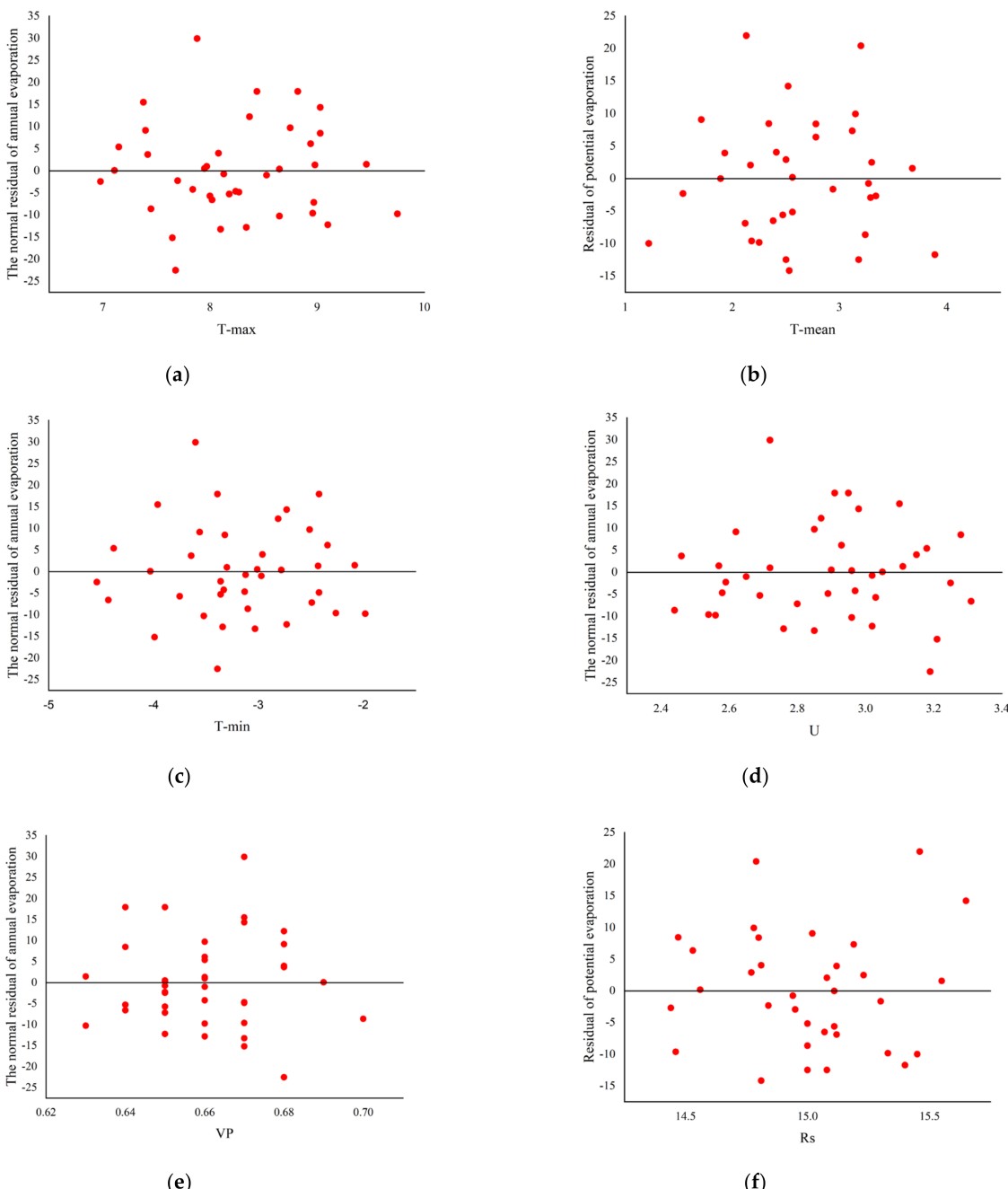

**Figure 10.** Residual difference between the meteorological factors and PET. (**a**) Maximum temperature; (**b**) mean temperature; (**c**) minimum temperature; (**d**) wind speed; (**e**) vapor pressure; (**f**) solar radiation.

## 5. Conclusions

This paper studied the trend of potential evaporation from 1961 to 2020 and the trend of climate change from 1984 to 2017 in Heilongjiang province. The potential evaporation shows an insignificant decline over time, which indicates that there is a certain evaporation paradox in this area. It decreased significantly in spring, but did not increase significantly in summer, autumn, or winter, indicating that the change in potential evaporation in spring had the greatest influence on the overall trend. From the perspective of location, the potential evaporation gradually increased from north to south, with the minimum value appearing in the Greater and Lesser Hinggan Mountains and the Mudanjiang area, and the maximum value appearing in the southern Sanjiang Plain area.

The temperature in Heilongjiang province is on the rise, which is in line with the background of global warming, while wind speed and solar radiation are on the decline. By using the multiple regression analysis method, it was calculated that the contribution rates of temperature, wind speed, relative humidity, and solar radiation to potential evaporation are $-87.23\%$, $88.25\%$, $65.87\%$ and $42.53\%$, respectively. Wind speed is the main reason for the low potential evaporation, followed by vapor pressure and temperature.

However, potential evapotranspiration can not completely represent the trend of evaporation change in the study area, and subsequent correlation analysis should be carried out based on the actual evapotranspiration of the local area, so as to verify the response relationship between climate change and evaporation.

**Author Contributions:** Conceptualization, Q.S.; methodology, Q.S. and Y.Z.; software, Q.S. and Y.Z.; validation, Q.Z., Q.S. and Y.Z.; formal analysis, Q.Z.; investigation, C.D.; resources, Q.Z.; data curation, Q.S.; writing—original draft preparation, Q.S.; writing—review and editing, Q.S.; visualization, Q.S.; supervision, C.D.; project administration, C.D.; funding acquisition, C.D. All authors have read and agreed to the published version of the manuscript.

**Funding:** Suppoted by Strategic Priority Research Program of the Chinese Academy of Sciences (Grant No. XDA28100105); Project of the National Natural Science Foundation of China (Water Resources and Hydropower Science Foundation).

**Institutional Review Board Statement:** Not applicable.

**Informed Consent Statement:** Not applicable.

**Data Availability Statement:** Data is not available due to privacy or ethical restrictions.

**Acknowledgments:** I would like to thank Changlei Dai for his guidance on my thesis. Thanks to Qingsong Zhang and Yang Zhou for their help in the data processing for this paper.

**Conflicts of Interest:** The authors declare no conflict of interest.

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
