# Peer review of "Analysis of Potential Evapotranspiration in Heilongjiang Province"

_sustainability, doi:10.3390/su152115374_

Round 1
Reviewer 1 Report
Most of the Figures must be modified (both maps and graphics).
Some sentences are just too large, this must be improved.
References need to be updated.
Conclusion is too poor
Author Response
Dear Reviewer:
Thank you for your letter and for the reviewers’ comments concerning our manuscript entitled“Analysis of Variation Trend and Influencing Factors of Potential Evapotranspiration in Heilongjiang Province”(sustainability-2621517).Those comments are all valuable and very helpful for revising and improving our paper. We have studied comments carefully and have made correction which we hope meet with approval. The main corrections in the paper and the responds to the reviewer's comments are as flowing:
â‘ The review's comment: Most of the Figures must be modified (both maps and graphics).
The authors' answer: Thanks very much for your comments; I have modified the format of the pictures in the article.
â‘¡The review's comment: References need to be updated.
The authors' answer: Thanks very much for your comments; I have revised the references in the article
â‘¢The review's comment: Conclusion is too poor
The authors' answer: Thanks very much for your comments; I have modified it.
Yours sincerely,
Quanchcong Su

Reviewer 2 Report
The manuscript Analysis of variation trend and influencing factors of potential evapotranspiration in Heilongjiang Province elaborates a brief regarding the decreasing of Potential evapotranspiration (PET) in accordance with vapor pressure, solar radiation, wind direction, etc., and might gain the interest of the scientific community. However, there are certain aspects to be cleared in the manuscript. General and specific comments are added in the same section. Please go through the comments.
1. The abstract section needs to be modified. There is a huge gap between Line No. 16 and Line no 17, where the authors mentioned the methodology right after the importance of the Heilongjiang Province. Also, the abstract lacks the objective and brief of the uniqueness of the manuscript.
2. Please change “This” in line no. 37.
3. Please mention what domestically and internationally refer to.
4. Check CO2 in line no 62.
5. What does Ep in line no 67 mean? Please add the full form if to be used for the first time.
6. Please elaborate on the last paragraph of the introduction with a bit more information on the uniqueness of the manuscript, and specific objectives. Also, add the hypothesis as to why this study will be of importance to study PET in other basins.
7. Please add the range of rainfall in the basin as this might be a useful parameter during discussion.
8. What does the background in Figure 1 refer to? What does -162 in the value at the legend refer to? If it’s a land use map being used, please reclassify the satellite image. Also, make the figure a bit large so that all the IDs of meteorological stations can be identified clearly. Also, change the scale from miles to km.
9. Check the spelling and grammar in Lines no 116, 117, 153, 155, 167.
10. What does NS in Figure 2 and in other figures refer to?
11. What is the error percentage of Kriging being used in Figure 7? Mention RMSE value.
12. The discussion seems a bit unclear. Please add some other references on the PET variation from other basins similar to the Heilongjiang Province.
Moderate editing on grammar is required.
Also spelling and punctuation, caps are to be checked thoroughly.
Author Response
Dear Reviewer:
Thank you for your letter and for the reviewers’ comments concerning our manuscript entitled“Analysis of Variation Trend and Influencing Factors of Potential Evapotranspiration in Heilongjiang Province”(sustainability-2621517).Those comments are all valuable and very helpful for revising and improving our paper. We have studied comments carefully and have made correction which we hope meet with approval. The main corrections in the paper and the responds to the reviewer's comments are as flowing:
â‘ The review's comment: The abstract section needs to be modified. There is a huge gap between Line No. 16 and Line no 17, where the authors mentioned the methodology right after the importance of the Heilongjiang Province. Also, the abstract lacks the objective and brief of the uniqueness of the manuscript.
The authors' answer: Thanks very much for your comments; I have revised the abstract part of the article and revised the key words
â‘¡The review's comment: Please change “This” in line no. 37.
The authors' answer: Thanks very much for your comments; I have modified it.
â‘¢The review's comment: Check CO2 in line no 62.
The authors' answer: Thanks very much for your comments; I have modified it.
â‘£The review's comment: What does Ep in line no 67 mean? Please add the full form if to be used for the first time.
The authors' answer: Thanks very much for your comments; Ep is introduced as evaporation in this article, which I have modified.
⑤The review's comment: Please elaborate on the last paragraph of the introduction with a bit more information on the uniqueness of the manuscript, and specific objectives. Also, add the hypothesis as to why this study will be of importance to study PET in other basins.
The authors' answer: Thanks very much for your comments; I have rewritten the last paragraph of the introduction
â‘¥The review's comment: What does the background in Figure 1 refer to? What does -162 in the value at the legend refer to? If it’s a land use map being used, please reclassify the satellite image. Also, make the figure a bit large so that all the IDs of meteorological stations can be identified clearly. Also, change the scale from miles to km.
The authors' answer: Thanks very much for your comments; Figure 1 refers to the relative position of Heilongjiang Province in China and the distribution of meteorological stations in the province, where VALUE refers to elevation (m) and the scale has been adjusted.
⑦The review's comment: What does NS in Figure 2 and in other figures refer to?
The authors' answer: Thanks very much for your comments; In the article, all NS in the article represents no significant increase, and ND represents no significant decrease.
â‘§The review's comment: What is the error percentage of Kriging being used in Figure 7? Mention RMSE value.
The authors' answer: Thanks very much for your comments; In the article, I have added the corresponding parameter values.
⑨The review's comment: The discussion seems a bit unclear. Please add some other references on the PET variation from other basins similar to the Heilongjiang Province.
The authors' answer: Thanks very much for your comments; In the article, I revised the discussion part, added the comparison between the results and the previous results, and analyzed the reasons
Yours sincerely,
Quanchcong Su

Reviewer 3 Report
Title: is too long and should be shortened "Analysis of Variation Trend and Influencing Factors of Potential Evapotranspiration in Heilongjiang Province" -> e.g. Analysis of Potential Evapotranspiration in Heilongjiang Province.
Abstract: your language and descriptions are vague in some sentences, e.g. 'The results indicated an overall declining fluctuation trend in PET in Heilongjiang Province, with a decrease of 7.776 mm/10a. Temperature, wind speed, vapor pressure, and solar radiation contribute -87.23%, 88.25%, 65.87%' - declining fluctuation trend is what period? over how many years? is this based on historic period analysis or projections?! you need to provide more detailed information on the study method, dataset, parameters monitored/measured, and the study period.
Keywords: I am not sure if this work investigates climate change, I know PET is changing due to changing climate, but you are not studying climate change. It is good to mention the case study location as a keyword.
Introduction: The early part of the intro section should set out the problem and the significance of addressing it. Although the problem statement is fine but I cannot see any discussion on the importance of the problem, i.e. the changing climate and the consequent fluctuations on temperature and radiations can change the balance in hydrological cycles including PET and therefore, alter the dynamic behaviour of ecosystems as already being documented in areas most vulnerable to climate change (10.1038/s41598-023-32343-8; 10.3390/hydrology10010016; j.jhydrol.2023.129229). I think its important to draw line between the climate change-induced fluctuations in PET and the 'consequences' such as alteration and degradation of sensitive ecosystems.
similarly, the consequences of paradox in PET is not mentioned, i.e. what happens if we have decline of PET should be described for the readers.
L76 - 'China's grain engineering' or grain production?
L86 '2. 2.Study area and data processing' - you need to remove the additional 2!
L87 'Heilongjiang Province is across the river from Russia' which river? mention the name.
Figure 1 needs additional sub-figure showing the map of china and neighbouring countries, highlighting the location of case study region. Current figure 1 is not very informative and doesnt provide the geographical context of the case study region and its land cover, geophysical features.
The dataset should be provided. currently, the link you provided i.e. 'https://data.tpdc.ac.cn/home' is a generic link and instead of this you need to put the link to the actual dataset used in this study.
Method: is brief but contain the necessary information. Perhaps referencing to detailed method papers could be helpful for the readers.
Results: Figure 2 should be replaced with a heatmap so that reader can better understand the trend and the extent of increase and decrease, the current figure is not appropriate.
Figure 3 need description and clarifications, i.e. is the PET reduction shown for a specific station or averaged over all stations? the axis title should be changed and make sure all the text is in english
Figure 4, your caption need to have description for (A) and (B) sub-figures.
The structure of the paper is quite weak, e.g. you have Figure 2, 3, 4, 5, and 6 exactly back to back without any text, description, and discussions of the results between them. This is not very good and you need to add detailed discussion of each figure.
another problem, your figures are not very consistent in formatting and your axis font size is too small to read. All figures need to be revised.
Another issue is the low resolution and quality of the images e.g. figure 6. This must be improved and enhanced in the revision.
Figure 7. 'Spatio-temporal variation of PET' -> the map is showing the spatial variation of PET, but i cannot see any temporal data? you need to clarify how this is addressing 'temporal' data?
Figure 8. is not useful in the present form, just saying increase or decrease is not important, instead you need to mention the extent of the increase and decrease.. i.e. by how much! this information need to be quantitative and not qualitative.
Conclusions need to mention the limitations of the study and direction for future research.
Proofreading and language checks are necessary. formatting inconsistency and use of weak grammar/ wording should be improved.
Author Response
Dear Reviewer:
Thank you for your letter and for the reviewers’ comments concerning our manuscript entitled“Analysis of Variation Trend and Influencing Factors of Potential Evapotranspiration in Heilongjiang Province”(sustainability-2621517).Those comments are all valuable and very helpful for revising and improving our paper. We have studied comments carefully and have made correction which we hope meet with approval. The main corrections in the paper and the responds to the reviewer's comments are as flowing:
â‘ The review's comment: Title: is too long and should be shortened "Analysis of Variation Trend and Influencing Factors of Potential Evapotranspiration in Heilongjiang Province" -> e.g. Analysis of Potential Evapotranspiration in Heilongjiang Province.
The authors' answer: Thanks very much for your comments; I have revised the title according to your suggestion.
â‘¡The review's comment: Abstract: your language and descriptions are vague in some sentences, e.g. 'The results indicated an overall declining fluctuation trend in PET in Heilongjiang Province, with a decrease of 7.776 mm/10a. Temperature, wind speed, vapor pressure, and solar radiation contribute -87.23%, 88.25%, 65.87%' - declining fluctuation trend is what period? over how many years? is this based on historic period analysis or projections?! you need to provide more detailed information on the study method, dataset, parameters monitored/measured, and the study period.
The authors' answer: Thanks very much for your comments; I have re-edited the abstract.
â‘¢The review's comment: Keywords: I am not sure if this work investigates climate change, I know PET is changing due to changing climate, but you are not studying climate change. It is good to mention the case study location as a keyword..
The authors' answer: Thanks very much for your comments; I have replaced the keywords.
â‘£The review's comment: L76 - 'China's grain engineering' or grain production?
L86 '2. 2.Study area and data processing' - you need to remove the additional 2!
L87 'Heilongjiang Province is across the river from Russia' which river? mention the name.
The authors' answer: Thanks very much for your comments; I have corrected the error information and missing information in the article
⑤The review's comment: Figure 7. 'Spatio-temporal variation of PET' -> the map is showing the spatial variation of PET, but i cannot see any temporal data? you need to clarify how this is addressing 'temporal' data?
The authors' answer: Thanks very much for your comments; I have changed the map name to spatial variation.
â‘¥The review's comment: Figure 1 needs additional sub-figure showing the map of china and neighbouring countries, highlighting the location of case study region. Current figure 1 is not very informative and doesnt provide the geographical context of the case study region and its land cover, geophysical features.
The dataset should be provided. currently, the link you provided i.e. 'https://data.tpdc.ac.cn/home' is a generic link and instead of this you need to put the link to the actual dataset used in this study and updated the detailed data website.
The authors' answer: Thanks very much for your comments; I have replaced Fig.1 with a detailed picture
⑦The review's comment: The structure of the paper is quite weak, e.g. you have Figure 2, 3, 4, 5, and 6 exactly back to back without any text, description, and discussions of the results between them. This is not very good and you need to add detailed discussion of each figure.
The authors' answer: Thanks very much for your comments; I've restructured the article
â‘§The review's comment: another problem, your figures are not very consistent in formatting and your axis font size is too small to read. All figures need to be revised.
Another issue is the low resolution and quality of the images e.g. figure 6. This must be improved and enhanced in the revision. Figure 4, your caption need to have description for (A) and (B) sub-figures.
The authors' answer: Thanks very much for your comments; I have improved the resolution of the pictures and unified the digital format in the pictures. Multiple pictures are marked with (a)(b)(c)
⑨The review's comment: Introduction: The early part of the intro section should set out the problem and the significance of addressing it. Although the problem statement is fine but I cannot see any discussion on the importance of the problem, i.e. the changing climate and the consequent fluctuations on temperature and radiations can change the balance in hydrological cycles including PET and therefore, alter the dynamic behaviour of ecosystems as already being documented in areas most vulnerable to climate change (10.1038/s41598-023-32343-8; 10.3390/hydrology10010016; j.jhydrol.2023.129229). I think its important to draw line between the climate change-induced fluctuations in PET and the 'consequences' such as alteration and degradation of sensitive ecosystems.
similarly, the consequences of paradox in PET is not mentioned, i.e. what happens if we have decline of PET should be described for the readers.
The authors' answer: Thanks very much for your comments; I referred to the article you mentioned and rewrote the last part of the introduction and the purpose of the research.
Yours sincerely,
Quanchcong Su

Round 2
Reviewer 3 Report
Authors have revised and improved the manuscript. While a good number of points mentioned previously is now addressed in the revised manuscript, there are some further points to address before the paper gets published. Please follow the comments below:
Introduction: as previously mentioned, discussion on the importance of the changing climate and the consequent fluctuations on temperature and radiations can change the balance in hydrological cycles including PET and therefore, alter the dynamic behaviour of ecosystems (j.jhydrol.2023.129229).. you said you addressed this already but I can see missing references that need adding.
2 - should be labelled as 'materials and methods'
please note that page 3 is left blank.
Figure 1 needs clarification - is there two sub-figures? if so, then please add (a) and (b) and separate caption for each figure.
Figure 2 is important but not very informative, i.e. we cant readily compare the magnitude of increase and decrease, to improve this you can plot heatmap of this data to allow for visual comparison. Similar issue is valid for figure 9 of the revised manuscript and need to be addressed.
conclusions: I cant see any discussion on the limitations of the data and method used, as well as direction for future research need.
language checks and proofreading is required.
Author Response
Dear Reviewer:
Thank you for your letter and for the reviewers’ comments concerning our manuscript entitled“Analysis of Variation Trend and Influencing Factors of Potential Evapotranspiration in Heilongjiang Province”(sustainability-2621517).Those comments are all valuable and very helpful for revising and improving our paper. We have studied comments carefully and have made correction which we hope meet with approval. The main corrections in the paper and the responds to the reviewer's comments are as flowing:
â‘ The review's comment: Introduction: as previously mentioned, discussion on the importance of the changing climate and the consequent fluctuations on temperature and radiations can change the balance in hydrological cycles including PET and therefore, alter the dynamic behaviour of ecosystems (j.jhydrol.2023.129229).. you said you addressed this already but I can see missing references that need adding.
The authors' answer: Thanks very much for your comments; According to your DOI number, I did not find the relevant literature, please send me the specific article name, I will add the article to the reference.
â‘¡The review's comment: I have changed Figure 2,9.
The authors' answer: Thanks very much for your comments; I have modified it.
â‘¢The review's comment: Figure 1 needs clarification - is there two sub-figures? if so, then please add (a) and (b) and separate caption for each figure.
The authors' answer: Thanks very much for your comments; Figure 1 does not have two sub-graphs.
â‘£The review's comment: conclusions: I cant see any discussion on the limitations of the data and method used, as well as direction for future research need.
The authors' answer: Thanks very much for your comments;I have added research limitations and future research directions in the conclusion.
Yours sincerely,
Quanchcong Su
Round 3
Reviewer 3 Report
Authors have revised the manuscript and addressed the remaining issues with expected level of details. There is only one outstanding issue that the authors asked, and I provided explanation below:
"The authors' answer: Thanks very much for your comments; According to your DOI number, I did not find the relevant literature, please send me the specific article name, I will add the article to the reference." - the doi link is https://doi.org/10.1016/j.jhydrol.2023.129229, please address accordingly.
â‘¡The review's comment: I have changed Figure 2,9. --> yes this is much better now.
â‘¢The review's comment: Figure 1 needs clarification - is there two sub-figures? if so, then please add (a) and (b) and separate caption for each figure. --> yes, this is now more clear.
â‘£The review's comment: conclusions: I cant see any discussion on the limitations of the data and method used, as well as direction for future research need. The authors' answer: Thanks very much for your comments;I have added research limitations and future research directions in the conclusion. --> yes, this is addressed accordingly.
overall acceptable level of writing but you need to proofread.
Author Response
Dear Reviewer:
Thank you for your letter and for the reviewers’ comments concerning our manuscript entitled“Analysis of Variation Trend and Influencing Factors of Potential Evapotranspiration in Heilongjiang Province”(sustainability-2621517).I have added this reference.
Yours sincerely,
Quanchcong Su
